# Growth and decay of the Iceland Ice Sheet through the last glacial cycle

Alexis Arturo Goffin <sup>1</sup>, Lev Tarasov<sup>1</sup>, Ívar Örn Benediktsson<sup>2</sup>, and Joseph M. Licciardi<sup>3</sup>

Correspondence: Alexis Arturo Goffin (aagoffin@mun.ca)

Abstract. Constraining the dynamic evolution of past ice sheets is critical for unravelling their responses to external forcing and feedbacks over long timescales. This is particularly true in the context of marine ice sheet collapse, as this is one of the largest sources of uncertainty for future sea-level rise projections. The Iceland Ice Sheet (IIS) provides an empirically constrained case study for investigating such an instability, having retreated from a predominantly marine-based ice sheet to isolated mountain ice caps during the last deglaciation. However, previous reconstructions of the IIS have been limited by either sparse data or a restricted exploration of model parameter space, lacking a robust quantification of uncertainties. Here, we address this gap by performing a truncated history matching of the last glacial cycle of the IIS. We use the Glacial Systems Model (GSM) constrained by a curated set of geochronological data to generate an envelope of plausible ice sheet histories.

Our results indicate that numerous asynchronous ice streams effectively drain ice from the interior to the margins, resulting in an extensive yet relatively thin ice sheet. During its local Last Glacial Maximum (23.6-20.9 ka), the IIS reaches the continental shelf edge in most sectors with a total volume of 0.41 to 0.76 metres equivalent sea level (m e.s.l.). In the most extreme, yet plausible, glaciation scenarios, our model reveals an ice bridge connecting the Iceland and Greenland ice over the Denmark Strait.

We find that accelerated ice discharge (at the grounding line) dominates mass loss during deglaciation. This acceleration is primarily driven by atmospheric warming through a cascade of mechanisms: surface meltwater induces hydrofracturing, leading to ice shelf disintegration, which in turn reduces buttressing and triggers rapid ice stream acceleration. The critical role of hydrofracturing in enabling model capture of deglacial data constraints is shown by an explicit sensitivity experiment. This thereby supports inclusion of hydrofracturing for modelling of ongoing ice sheet response to climate change.

## 20 1 Introduction

Understanding marine ice sheet potential for collapse is crucial for projecting future sea-level rise (Robel et al., 2019). However, this large-scale instability, not yet observed in the modern era, and its associated drivers remain poorly known (Robel et al.,

<sup>&</sup>lt;sup>1</sup>Department of Physics and Physical Oceanography, Memorial University of Newfoundland, St. John's, NL, Canada

<sup>&</sup>lt;sup>2</sup>Institute of Earth Sciences, University of Iceland, Reykjavík, Iceland

<sup>&</sup>lt;sup>3</sup>Department of Earth Sciences, University of New Hampshire, Durham, USA

2019; Coulon et al., 2024; Pattyn and Morlighem, 2020; Robel et al., 2019). The Iceland Ice Sheet (IIS) was a predominantly marine-based ice sheet with multiple ice streams during the last glacial cycle that ended with near complete deglaciation by the mid-Holocene (Patton et al., 2017; Benediktsson et al., 2022b). It thereby offers an empirically constrained case study to investigate such a termination.

Reconstructing the IIS evolution is challenging due to large uncertainties associated with: (i) oceanic and atmospheric conditions with strong dependence on the Atlantic Meridional Overturning Circulation (AMOC) (Xiao et al., 2017); (ii) controls on grounding line stability and migration; (iii) calving and sub-shelf melt processes; and (iv) ice stream drainage tied to the strong geothermal activity (Flóvenz and Saemundsson, 1993; Hjartarson, 2015). The warm/cold oscillations were controlled by the variability of the cold East Icelandic Current and the relatively warm Irminger Current derived from the North Atlantic Current (Xiao et al., 2017). The local climate was therefore contingent on AMOC strength. Based on dated marine limit shorelines, Norðdahl and Ingólfsson (2015) suggest that the marine-based part of the ice sheet collapsed during the Bølling–Allerød, emphasizing the key role of calving and bedrock topography in the deglaciation. The tectonic and magmatic legacy of Iceland translates to a strong geothermal heat flux (Flóvenz and Saemundsson, 1993; Hjartarson, 2015), conducive to basal melting. Geomorphologists identified streamlined subglacial bedforms, suggesting the presence of fast-flowing paleo ice streams (Bourgeois et al., 2000; Spagnolo and Clark, 2009; Principato et al., 2016; Benediktsson et al., 2022b, a) . Digital elevation models reveal troughs terminating at moraines (Fig. 1) (GEBCO Bathymetric Compilation Group 2023, 2023), with the troughs likely carved by ice streams during past glaciations (Spagnolo and Clark, 2009; Benediktsson et al., 2022a).

Despite recognition of these physical uncertainties, empirical studies alone cannot fully quantify such uncertainties or capture potential nonlinearities and synergistic effects of multiple forcings acting in concert. Moreover, proxy records and glaciological imprints from mapping have enabled reconstructions of past IIS margins (Spagnolo and Clark, 2009; Benediktsson et al., 2022b, 2023a, b, c), these reconstructions lack confidence for time intervals and areas with sparse data coverage. Consequently, significant knowledge gaps persist in this empirical framework particularly for the pre-Last Glacial Maximum (LGM) configuration, the southern and eastern margins, the ice volume and sea level contribution during the LGM, as well as the drivers and controls of the deglaciation (Benediktsson et al., 2022b, 2023a).

Numerical models offer an effective tool to bridge these gaps through glaciologically self-consistent approaches. Previous modelling studies (Hubbard et al., 2006; Hubbard, 2006; Patton et al., 2017) have significantly advanced our understanding of IIS dynamics, simulating a relatively fast ice sheet with a low aspect ratio and strong sensitivity to atmospheric temperature and geothermal conditions. However, these studies explored only limited portions of the parameter space, typically hand-tuning a small number of model parameters (

Figure 1. Bathymetric and topographic map of Iceland from GEBCO DEM with the present-day glacial situation (Farinotti et al., 2019).

Systems Model (GSM, Tarasov et al., 2025) configured with 35 ensemble parameters to partially account for uncertainties in climate, basal drag, and marine ice processes.

Our results address several key research questions about the IIS. We investigate the configuration, dynamics and evolution of the IIS throughout the last glacial cycle (Sect. 3.2). We then disentangle the drivers and controls of its subsequent deglaciation 3.2.3).

## 2 Methods

We describe below the glaciological model and our methodology for constraining it against paleo data.

## 65 2.1 The ice sheet model

The Glacial Systems Model (GSM) is a 3-D thermo-mechanically coupled hybrid Shallow Ice Approximation (SIA) / Shallow Shelf Approximation (SSA) ice-sheet model that resolves ice streams, ice shelves, and grounding line migration. The GSM includes a permafrost resolving bed-thermal model. It incorporates fully coupled visco-elastic glacio-isostatic adjustment that accounts for the changing load of the adjacent Greenland Ice Sheet. The GSM uses a combined positive degree day (PDD) and

**Figure 2.** Flow chart illustrating the main steps to calibrate paleo-ice sheets. Blues boxes indicate the model processes, orange boxes represent the constraint data, gray boxes denote the history matching loop. Rounded shapes indicate the function while squared shapes the input/output.

٠

**Figure 3.** Glacial indices derived from smoothed temperature anomalies (50 years running mean, interpolated to 50 year time steps) from the NGRIP ice core (nor, 2004) and seasonal variations from the LCice model (Geng et al., 2025) (February, April, and July). Raw signals are shown in translucent.

novel positive temperature short-wave parameterization to account for the direct impact of changing orbital forcing on surface melt. The GSM also incorporates the marine ice cliff instability for marginal ice as well as hydro-fracturing. A complete description of the GSM is available in Tarasov et al. (2025).

We use a lat-lon grid with resolution of  $0.0625^{\circ}$  x  $0.125^{\circ}$  ( $\sim 7$  x 6 km). Our choice of grid resolution represents a trade-off between the need to resolve key physical processes and computational efficiency. The chosen grid resolution explicitly resolves the major ice streams, while keeping the computational cost manageable for running a large ensemble of simulations.

For climate forcing, the GSM uses a combination of an asynchronously coupled 2D energy balance climate model (EBM) and both yearly and monthly resolved glacial indices to interpolate between LGM monthly temperature, precipitation and evaporation climatologies (Paleo-Modelling Intercomparison Project 3, Braconnot et al., 2012) and present-day climatologies (ECMWF ERA5 reanalysis, Melchior Van Wessem et al., 2018). We use the North Greenland Ice Core Project (NGRIP) ice core record (nor, 2004) along with transient simulation output from a climate model (Geng et al., 2025) (Fig. 3) for the glacial indices.

Our model is forced by the ocean temperature field of the TraCE-21ka deglacial simulation (Liu et al., 2009) performed with version 3 of the Community Climate System Model (CCSSM3). Prior to 22 ka, the ocean forcing is extrapolated from the TRACE chronology using a glacial index, similarly to the atmospheric forcing. To partly address uncertainties in the ocean forcing, an ensemble parameter is applied to the index.

The bedrock topography is derived by subtracting the observed ice thickness field (Farinotti et al., 2019) from the GEBCO surface elevation DEM (GEBCO Bathymetric Compilation Group 2023, 2023).

https://doi.org/10.5194/egusphere-2025-5319 Preprint. Discussion started: 10 November 2025

© Author(s) 2025. CC BY 4.0 License.

Hank and Tarasov (2024) have demonstrated that the surge pattern over the Hudson Strait is very sensitive to the geothermal heat flux. However the 4 km deep geothermal heat flux under Iceland is not well-constrained. For this work, two geothermal heat flux fields are subject to weighted average according to an ensemble parameter. The two fields are generated by interpolating boreholes from Flóvenz and Saemundsson (1993) and Hjartarson (2015).

Our sediment distribution is the result of two merged fields: the NOAA offshore dataset (Straume et al., 2019) and a parameterized terrestrial field with 200 m of sediment at slope divergences and 0 m at convergences in high-resolution bedrock topography. We upscale the latter to the GSM resolution to preserve the high-resolution information. Seismic surveys indicate thick sediment layers in the valleys, likely deposited over the course of previous glaciations (Black et al., 2004).

#### 2.2 Paleo-constraints

To evaluate model plausibility, we compiled a comprehensive IcelandICE database of geological constraints that capture the spatial and temporal evolution of the IIS throughout the last glacial cycle (orange boxes in Fig. 2). Our database is a compilation and refinement of published geochronological data (Norddahl and Pétursson, 2005; Licciardi et al., 2007; Benediktsson et al., 2022b, 2023a, b, c), supplemented by geomorphological evidence (Spagnolo and Clark, 2009; Benediktsson et al., 2022b, a) (Fig. 4). The database is provided in the supplement.

The geochronological constraints are classified into two categories. One category is observations of past ice extent (paleoEXT) which consists of marine radiocarbon ages. The other is past ice thickness (paleoH) which consists of terrestrial cosmogenic and radiocarbon ages (Fig. 4). The radiocarbon ages are calibrated to calendar years. We introduce a multi-tier data quality assessment. Only data with rigorous age calibration, clear interpretation, and providing clear constraint value are assigned to tier 1. Most of the past ice thickness data was assigned to tier 1. However, outliers (i.e., ages that are anomalously anti-phasing with other surrounding data) were systematically ranged into tier 4 (e.g. Sandfell cosmogenic age which is too old).

Data with robust calibration ages and high constraining value but subject to widely different, yet justified, interpretations were classified as tier 1A. These data were explicitly tested for impact of constraint on the simulations within the context of the interpretative debate. Tier-2 data supplement Tier 1 data, providing finer details of the deglacial ice-sheet-thinning history. Tier-3 data comprise lower-quality constraints, marked by low-confidence in uncertainty specifications and ambiguous interpretation.

Past ice extent (paleoEXT) constraints provide ages for the deglaciation of marine ice. Such data are from marine sedimentary cores and require a strong understanding of the stratigraphic units. For instance, some age may reflect the mixing with older
sediments introducing ambiguity in the interpretation. Therefore, only data with a clear stratigraphic history are classified as
tier 1. Since most of the IIS was marine-based at the LGM, tier-1 marine data play the most critical role in constraining the key
deglacial changes. We compile the main radiocarbon ages from Benediktsson et al. (2022b, 2023a, b, c) while ensuring that the
dates from the source studies have acceptable uncertainties. In the Reykjafjarðaráll-Húnaflóadjúp trough on the Northern shelf
(~ 20.2°W, 66.9°N), a date of 15.0 ± 0.6 ka from post-glacial muds overlying a diamicton constrains the timing of ice retreat
from the trough (Andrews and Helgadóttir, 2003). On the western shelf, a deglaciation age of 15.3 ± 0.9 ka in the Jökuldjúp

125

130

135

155

trough [24.21°W, 64.29°N] is based on an extrapolation down to ice-contact sediments from three radiocarbon dates in the overlying marine mud (Jennings, 2000). Both of these data are classified as tier 1 and provide the most powerful constraint for the ice sheet retreat over the continental shelf. Tier 1A paleoExt data introduce important complexities and highlight areas under debate. For instance, in the Djupall trough on the Northwest shelf ( $\sim 24.0^{\circ}$ W,  $66.6^{\circ}$ N), radiocarbon dates (calibrated) of  $22.3 \pm 1.0$  ka and  $36.5 \pm 1.5$  ka suggest that glacier ice was absent during the LGM, and that the area was potentially occupied by shorefast sea ice instead (Geirsdóttir et al., 2002; Andrews et al., 2018). This is in contrast with geomorphological evidence suggesting more extensive ice cover. Furthermore, older Tier 1A dates from the Reykjafjarðaráll-Húnaflóadjúp trough indicate that ice had culminated there before 28 ka and persisted until the onset of deglaciation around 15 ka (Andrews et al., 2000).

Past ice thickness is constrained by cosmogenic nuclide exposure ages from mountain summits, nunataks, as well as radio-carbon ages from sites that constrain local deglaciation timing (paleoH). We compiled and recalculated the cosmogenic  $^3$ He ages of table mountain summits from Licciardi et al. (2007) using version 3 of the online exposure age calculator formerly known as the CRONUS-Earth calculator (Balco et al., 2008). We use the same locally calibrated Iceland  $^3$ He production rate from Licciardi et al. (2006), time-varying geomagnetic scaling LSDn (Lifton et al., 2014), and the same isostatic adjustments applied in Licciardi et al. (2006). Ages are calculated with no erosion, as the erosion rate is poorly quantified. No adjustments are made for snow cover, which if applied would result in older exposure ages for the table mountain summits, but the time-integrated local snow history at the sample sites is poorly known. The recalculated table mountain  $^3$ He ages are within  $\sim 1$ % of the ages originally reported in Licciardi et al. (2007). Key Tier 1 paleoH constraints include cosmogenic  $^3$ He ages from several table mountains such as Herðubreið (10.4  $\pm$  1.0 ka, 1522 m), Búrfell (10.8  $\pm$  0.6 ka, 949 m), and Bláfjall (14.3  $\pm$  0.8 ka, 1087 m) (Licciardi et al., 2007). These sites provide crucial lower limits on ice thickness in the interior of Iceland. Additionally, radiocarbon dates from marine shells, such as those from Vopnafjörður (11.0  $\pm$  0.3 ka, 50 m) and Hvalvík (14.7  $\pm$  0.4 ka, 40 m), constrain the timing of local deglaciation.

While not directly dated, extensive geomorphological evidence provides further context for understanding the IIS configuration and dynamics. Streamlined subglacial ridges and cross-shelf bathymetric troughs indicate the presence of fast-flowing ice streams that drained the ice sheet interior (Spagnolo and Clark, 2009; Principato et al., 2016; Benediktsson et al., 2022a). Terminal moraines identified at or near the continental shelf edge in most sectors support an extensive LGM ice sheet (Patton et al., 2017). Additionally, the radial pattern of these landforms suggests a primary ice divide near the center of Iceland (Spagnolo and Clark, 2009).

The spatial distribution of the geological constraints is uneven. As shown in Fig. 4, data are scarce on the southern and eastern regions of the Icelandic continental shelf.

## 2.3 Experimental design

We produced a large ensemble of simulations based on Latin Hypercube sampling of parameter vectors to efficiently explore the model space, ensuring broad coverage of parameter interactions (blue boxes in Fig. 2). Each of these simulations is characterized by a parameter vector that specifies the value of all ensemble parameters. Sensitivity tests were used to identify which potential GSM ensemble parameters had significant impact on various model run statistics. As a result of these tests, the GSM

**Figure 4.** Our Iceland constraint database summary plot. The topographic map was generated using GEBCO DEM and the geomorphological imprints of the IIS on the Iceland shelf are based on Spagnolo and Clark (2009), Benediktsson et al. (2022b), and Benediktsson et al. (2022a)

was configured with 35 ensemble parameters (9 for ice dynamical/basal drag, 15 for climate forcing and surface mass balance, 5 for ice calving/subshelf melt, 1 for deep geothermal heat flux, and 2 for basal hydrology). Each simulation is initialized with the present-day ice thickness field (Farinotti et al., 2019) and runs from 122 ka to 2000 CE. Each simulation was then evaluated against the data (see History Matching in the next Sect. 2.4).

In addition, we conducted sensitivity experiments to further decipher the role of potential key processes contributing to past IIS variability. Specifically, we isolated the role of surface crevassing on ice shelf degradation through an experiment in which we turn-off hydrofracturing (parameterized as per (Pollard et al., 2015)), restarting the non ruled out yet (NROY) runs from 15 ka to the present day. We also conducted a similar experiment with sea level held constant.

Overall, in this study, we conducted a total of  $\sim 7000$  simulations: 6000 for ensembles, and 1000 for sensitivity experiments.

## 165 2.4 History Matching

160

History matching identifies a set of model chronologies that are not ruled out given available data constraints and robust uncertainty analysis (gray boxes in Fig. 2). As such, it aims to "bracket reality" as opposed to the much more difficult task of

180

190

determining a meaningful most likely chronology (Tarasov and Goldstein, 2021). Each simulation is scored against key metrics and subsequently deemed implausible or not via multiple sievings. To be meaningful, this must account for both model and data uncertainties. The resulting products are sets of NROY (not ruled out yet) chronologies.

The scoring uses all tier 1, 1A and 2 constraint data (past ice thickness score and past ice extent score). This implausibility metric is a conceptual distance between the model output and the associated data constraint (see equation below). The past ice thickness score is calculated by comparing the paleoH observation to the corresponding model grid cell ( $\pm$  spatial uncertainty =  $\approx 5$  km) and time slice ( $\pm$  time uncertainty =  $\approx 5$  the used a similar approach for the tier 1A past ice marine extent constraint, where the score is based on whether the model grid cell was covered by grounded ice, floating ice, or was ice-free, depending on the sample type. The samples can either infer the presence of open marine conditions (OMC), sub-ice shelf (SIS), or proximal to grounding line (PGL). As per the two tier 1 past marine extent constraints, the past deglaciation score is calculated by evaluating the capability of the model to expand over the data points during Marine Isotope Stage 2 (MIS 2) and subsequently deglaciated within the calibrated  $2\sigma$  range data ages. The implausibility values  $I_i$  for each datum and each parameter vector is calculated as follows:

$$I_i^2(x,cm) = \frac{\left(M_i(x,cm) - d_i(x) - \epsilon_{\text{total}}\right)^2}{\sigma_{struct}^2 + \sigma_{obs}^2}$$

where  $M_i(x,cm)$  is the model output at location and time x with parameter vector cm,  $d_i(x)$  the datum value,  $\epsilon_{\text{total}}$  the mean bias error term, and  $\sigma_{obs}$  the structural and observational standard deviations respectively.

In this study, we applied three different sievings to produce 3 distinct subsets.

The first and most critical sieving consists of ruling out simulations if they are deemed implausible according to the top tier-1 data. In particular, we ruled out simulations if:

- the implausibility with respect to the two top tier marine constraints is above 3, a rejection threshold based on the 3  $\sigma$  rule (Pukelsheim, 1994). The two marine constraints are the datum at [24.21°W, 64.29°N] (Andrews et al., 2000) and the datum at [20.23°W, 68.86°N] (Jennings, 2000; Benediktsson et al., 2022b), with inferred  $2\sigma$  deglaciation time intervals of [13.52-16.51] ka and [13.41-17.16] ka respectively.
- the implausibility with respect to the top tier 1 past ice thickness data is above 3.

The resulting NROY $_{tier1}$  subset is the main product of the present study that is used to bracket the last glacial cycle evolution and serves as the main basis for our analysis (see Sect. 3.2).

The second sieving provides an IC<sub>data\_tier1</sub> subset based on the same sieving as NROY<sub>tier1</sub> but without accounting for the structural uncertainty. This gives a narrower set of chronologies that are nominally consistent with the data. The third sieving consists of ruling out simulations deemed implausible by considering both, the tier-1 and tier-1A data, within model and data uncertainty. The same rejection threshold of 3 was applied. Here, the resulting product is a set of NROY chronologies: NROY<sub>tier1-1A</sub>.

Figure 5. Past ice extent (a) and past ice thickness (b) misfit scores. The data ID locations are shown in Fig. A1.

## 2.5 Ice discharge assessment

We used two different approaches to estimate the mass of ice discharged into the ocean. Our first approach calculates the ice discharge at the grounding line using the component method: we subtract the total net mass balance from the surface mass balance of grounded ice, D = SMB - MB. The second approach calculates the ice discharge over gates placed within bathymetric troughs and valleys. That is, where ice streams are likely to appear. A similar approach was used by Stokes et al. (2016) for the Laurentide Ice Sheet.

## 205 3 Results

This results section first evaluates model performance against geochronological constraints (Sect. 3.1), then presents the bracketed chronology of IIS evolution through the last glacial cycle based on the main NROY subset (Sect. 3.2).

## 3.1 Model fits to data constraints and scoring

For this study, we conducted a total of  $\sim 7000$  simulations. Here the 2000 final large ensemble members are compared to the tier-1 and tier-1A data using  $3\sigma$  implausibility thresholds.

The NROY<sub>tier1</sub> (tier 1) subset has 107 members. The  $IC_{data\text{-tier1}}$  subset (only constraint data uncertainties) has 44 members. And the NROY<sub>tier1-1A</sub> subset (including the contradictory 1A constraint data) has only 23 members.

While there is a relatively low misfit score with respect to paleoH data (most NROY<sub>tier1</sub> scores  $



#### 3.1.1 Past marine ice extent

The marine ice extent data provide critical constraints for evaluating model performance across Iceland's continental shelf sectors. Here we focus on the data-model comparison for the two NROY subsets NROY<sub>tier1</sub> and NROY<sub>tier1-1A</sub>. Each of the NROY members are compared against the tier 1 and tier 1A past ice extent data which can infer proximal-to-GL (PGL) conditions, sub-ice shelf (SIS) conditions, or open marine conditions (OMC). Overall, while the tier 1 observations are bracketed by the NROY sub-ensembles (site 2003,2006), tier-1A data fitting presents a greater challenge (2002,2004) (Fig. 5a).

In the Reykjafjarðaráll-Húnaflóadjúp trough located on the Northern shelf ( $\sim 20.2^{\circ}$ W, 66.9°N), the NROY sub-ensembles perform well with the tier-1 observation (2003) but most of the NROY members struggle to bracket the oldest datum (2002). The latter PGL constraint (2002) suggests ice free conditions at 28.6 ka, whereas most NROY simulations indicate sustained glaciation around 30 ka or earlier.

In the Northwest Djúpáll trough ( $\sim 24.0^{\circ}\text{W}$ , 66.6°N), none of the NROY members bracket the data point (2004), suggesting that the trough was ice free at  $\sim 22.3$  ka. However, the Djúpáll data are subject to methodological limitations including core depth, the number and location of cores, datable material, reservoir corrections, and interpretations of different stratigraphic units, all of which introduce ambiguity. Andrews et al. (2002) claim that the exact location of the LGM margin is uncertain due to insufficient data, but that the Djúpáll trough was ice-free from 36 ka based on dated foraminifera in diamict and ice-rafted debris (IRD) in overlying sediment. Subsequent studies (Geirsdóttir, 2004; Chesley, 2005; Principato et al., 2006; Quillmann et al., 2012) propose revising this chronology, indicating that Djúpáll trough was most likely ice-filled during the LGM. The older dates of  $\sim 36$  ka may reflect reworking of older sediments rather than indicating a restricted LGM extent in Djúpáll. Consequently, more recent studies depict the ice sheet margin at the continental shelf edge at the LGM (e.g. Geirsdóttir et al., 2009; Geirsdóttir, 2011; Andrews et al., 2018).

Moreover, this conflicts with multiple nearby observations (e.g., SITID 2003, 2006, and 2007), which indicate persistent grounded ice across the continental shelf from  $\sim 25-15$  ka. The scenario implied by this data (2004), with the Djúpáll trough ice-free by  $\sim 22.3$  ka while grounded ice persisting further south in the Jökuldjúp trough [24.21°W, 64.29°N] until  $\sim 15.3 \pm 0.9$  ka (Jennings, 2000) [SITID = 2006], is glaciologically dubious.

On the western shelf, in the Jökuldjúp trough (2006), the NROY models, particularly the NROY<sub>tier1-1A</sub> subset, perform relatively well. The larger model-data mismatch for some NROY<sub>tier1</sub> simulations reflects the large spread in modelled grounding-line retreat within the Jökuldjúp trough (see details in Sect. 3.2). This spread is a direct result of parameter variations within the NROY<sub>tier1</sub> ensemble. In particular, the highly non-linear grounding-line dynamics in this deep marine embayment are very sensitive to parameters controlling basal sliding and calving rates. Consequently, while the NROY<sub>tier1-1A</sub> sub-ensemble captures the overall deglacial pattern, individual simulations can diverge from the timing of the constraint.

## 3.1.2 Past ice thickness

Cosmogenic exposure ages, and to a lesser extent radiocarbon ages, constrain past ice thickness enabling robust comparison between modelled and empirical ice sheet thinning (Fig. 5b). We evaluate  $NROY_{tier1}$  and  $NROY_{tier1-1A}$  ensembles against each






tier-1 past ice thickness observation using two complementary scoring approaches. First, "presence of ice" scoring gives zero misfit if modelled grid-cell ice thickness exceeds the empirical value prior to the data age. Second, "absence of ice" scoring gives zero misfit if modelled ice thickness is below the inferred ice thickness at the data age (Fig. A2). Both misfit scores increase proportionally with temporal offset between modelled and empirical values.

While the NROY subsets successfully bracket all paleo ice thickness observations, certain NROY simulations yield higher misfit scores for specific data points. For example, at Herðubreið (SITID = 1005), the empirical age of deglaciation ranges between 11.4 and 9.4 ka, but in one of the NROY simulations (run identification number nn555), deglaciation occurs earlier at 15 ka, resulting in a high misfit score for the "presence of ice" (score = 19). However, given the early deglaciation, the "absence of ice" condition after 9.4 ka is satisfied and the associated score is zero. Conversely, further north at Bláfjall (SITID = 1009), the empirical deglaciation window is 15.1-13.5 ka, whereas in one of the NROY simulations (nn599), deglaciation occurs later, at 10 ka. This leads to a low misfit score for the "presence of ice" but a high misfit (score = 36) for the "absence of ice", as the model grid cell should not have ice after 13.5 ka. Persistent data-model mismatches are likely in good part attributable to inadequate grid resolution for resolving the complex topographies of Iceland.

## 3.2 Last glacial cycle IIS chronology

The evolution of the IIS volume and area through the last glacial cycle for the full initial ensemble and more-data constrained NROY<sub>tier1</sub> sub-ensemble are presented in Figures 6a and A3. Out of 2000 ensemble members, 107 are NROY. Ensemble variance is not well correlated with ice volume (Fig. A4). However, once the ice sheet expands beyond the present-day coastline (i.e., once the ice area threshold of  $\sim 100 \times 10^3 \ km^2$  is crossed), the ensemble variance grows significantly, reflecting regionally divergent rates of grounded ice advance (Fig. 6b). Depending on the ensemble parameter vector choice, the grounding line fluctuates considerably in light of the gentle shelf slope. Furthermore, the relative variance, i.e., the variance normalized by the mean ice volume, peaks during the deglaciation (Fig. A4).

## 270 **3.2.1 pre-LGM**

For the Last Interglacial spin-up state, we initialize the model with the present-day ice thickness field (Farinotti et al., 2019). At the onset of the last glacial cycle, ice in our NROY<sub>tier1</sub> ensemble expands from two main centers: the eastern part of the central highlands from the existing ice caps, particularly Vatnajokull and, a smaller glaciation over the Northwest peninsula (Fig. 6b). By  $\sim 90$  ka the two glaciated ice bodies merge into an ice sheet covering the entire terrestrial landmass. The ice sheet is characterized by a double dome configuration, established on the foundations of the initial two ice bodies. Subsequently, at  $\sim 83$  ka, a rapid retreat separates the two domes.

Between MIS 5a and the end of MIS 3 (80-30 ka), the grounding line advances further and fluctuates across the continental shelf, constrained at its maximum by the shelf edge. Within the NROY<sub>tier1</sub>sub-ensemble, the main variations of extent occurs over the Northern and Southeastern continental shelf. For similar ice volumes, the direction of ice expansion can vary significantly, contingent on the parameter choice. For some parameter vectors, the absence of ice over the northern shelf (70 ka in Fig. 6b), can be attributed to the depth of bathymetric troughs (Fig. 1). Conversely, simulations show that grounded ice can

**Figure 6.** (a) Time series of IIS volume during the last glacial cycle. Initial full ensemble (gray shaded area) and sieved non-ruled out yet (NROY<sub>tier1</sub>) sub-ensemble (blue distribution) with median NROY<sub>tier1</sub> (black curve). (b) Density distribution of grounded ice within the NROY<sub>tier1</sub> sub-ensemble at different time slices.





extend across this area when anchored on shallow parts of the continental shelf such as the Kolbeinsey Ridge ( $\sim 19^{\circ}W, 67^{\circ}N$ ), part of the Mid-Atlantic Ridge. Furthermore, the Kolbeinsey Ridge can also act as a pinning point that buttresses and stabilizes the ice shelf, thereby affecting grounding line dynamics.

Owing to the underlying soft till, numerous troughs, and high geothermal heat flux, the ice sheet develops multiple ice streams across the continental shelf. Most of these ice streams activate and deactivate independently (Fig. A6), with basal velocities periodically dropping to zero, indicating complete shutdowns (Fig. A5a,d-f). Given this independence and the relatively stable, cold glacial climate (with insufficient oceanic or atmospheric warmth to induce substantial surface or sub-shelf melt (Fig. A7e), and thus little change in ice shelf buttressing) we infer that ice-stream activation is primarily driven by internal thermodynamic "binge-purge" cycling rather than external forcing. However, given the model's coarse resolution, other mechanisms cannot be ruled out. Preferential drainage pathways from the ice-sheet interior to the margin vary between simulations and through time within individual simulations, often favouring one trough over adjacent ones. In contrast, some ice streams remain continuously active in the Southwestern and northern shelf sectors (Fig. A5b,c).

The main limitation predating the LGM is lack of constraints. Consequently, no scenarios of ice expansion pathway can be ruled out and the ice margin position remains highly uncertain.

## 3.2.2 The Last Glacial Maximum

At MIS 2, the IIS advances further onto the shelf, specifically along the western margin. The ice sheet maintains its double-dome structure, with the primary dome centred over the present-day Vatnajokull region and a secondary dome over the North-west peninsula (Fig. 7). The simulated NROY<sub>tier1</sub> ice volume spans from 0.76 to 0.41 m eustatic sea level equivalent (mESL), with a median of 0.6 mESL. at the local LGM (23.6 to 20.9 ka). The large variance in LGM volume is primarily due to differences in the extent of the grounding line across the Southwestern shelf, the potential link with Greenland as well as variations in the overall ice thickness.

During the global LGM, summer air temperatures vary within our  $NROY_{tier1}$  sub-ensemble, but remain below freezing point, with little to no surface melt (Fig. A7e). The IIS expansion is therefore mainly constrained by the continental shelf edge and reaches it in most sectors. This is supported by the presence of numerous moraines all along the shelf edge (Fig. 4). The exception is in the southwest where the grounding line position is more variable across NROY models (Fig. 6b).

The ice sheet is bounded by ice shelves, particularly along the northern margin. Some NROY<sub>tier1</sub> simulations exhibit an ice bridge over Denmark Strait connecting the Northeastern Iceland ice shelf to the Greenland Ice Sheet (Fig. 7). The latter reached the shelf edge over its Southeastern margin during the LGM (Andrews et al., 2000; Dowdeswell et al., 2010; Funder et al., 2011; Vasskog et al., 2015). We have been unable to find definitive paleo data in either support or refutation of such an ice bridge. Even though the paleoceanographic data of the northern North Atlantic at the LGM reflect relatively warm conditions with perennial sea-ice cover confined to the northeast Greenland margin area (Kucera et al., 2005), there is evidence for a reduction in the strength of the Western Boundary undercurrent (WBUC) around Southern Greenland during glacial stages (Müller-Michaelis and Uenzelmann-Neben, 2014). This is likely due to reduced Denmark Strait Overflow (Andrews et al.,






2018), potentially attributed to the presence of an ice shelf. In some simulations, grounded ice extends fully across Denmark Strait (Fig. 6b).

The model simulates thicker ice when ensemble parameters enhance surface accumulation and basal drag. Increased basal drag reduces ice velocity, consequently reducing mass transport to ablation areas.

The presence of a thick, soft basal sediment layer in marine sectors and valley bottoms as well as high geothermal heat flux facilitate fast basal sliding. This partly explains why all simulations, regardless of the parameter choice, depict a highly dynamic ice sheet with numerous fast-flowing ice streams that effectively drain the ice from the interior to the margin (Fig. 8a). Another factor are the numerous topographic troughs that over time guide ice stream drainage of the ice sheet (Fig. 1 and 8b). Most if not all of these troughs are likely the result of the two-way feedback between topographic excavation by subglacial erosion and topographic steering and confinement of ice streaming. However, at any given time, there are varying degrees of topographic steering and confinement with some ice streams clearly topographically confined, and others crossing over topographic ridges (Fig. 8b).

Given the temporal variation in ice stream configuration for a given simulation as well as the variation between ensemble members (Fig. A6), no single simulation at a given time is likely to be fully consistent with relevant geological data such as streamline ridges (Fig. 8a). A further challenge in such comparison is the poor age control on such features, with only indirect inferences that they were formed during the local LGM (Hoppe, 1968; Bourgeois et al., 2000; Spagnolo and Clark, 2009; Principato et al., 2016; Benediktsson et al., 2022a).

Such effective drainage results in an extensive ice sheet with a small mean ice thickness on the order of  $811 \pm 71$  m as per the NROY sub-ensemble. That is, lower than crude estimates based on volume-area scaling (using a scaling exponent of 1.23 from (Paterson, 1994) and an area of 250 000  $km^2$ ; mean ice thickness = 1026 m), and even lower than past models for Iceland (939 m for Hubbard et al. (2006) and 1172 m for Patton et al. (2017)).

## 3.2.3 The last deglaciation

The IIS loses  $\sim 95$  % of its mass during the last glacial-interglacial transition ( $\sim 20.9$  to  $\sim 9$  ka). The deglacial mass loss is dominated by calving until  $\sim 11.5$  ka. The IIS deglaciation history is characterized by a two-step evolution with five distinct phases: three of moderate ice retreat or even re-advance (I, III, and V in Fig. 9), and two of intervals of rapid ice loss (II and IV in Fig. 9).

First, between the end of the local LGM and 14.6 ka (Phase I), the grounding line retreats moderately across the Southwestern shelf (Fig. 6b). The mass loss is tied to a slight offset between ice discharge and surface mass balance (Fig. 9d). This is driven by sea level rise alone (Fig. 9a) as the climate is still relatively cold (Fig. 9b, c), consistent with TraCE-21000 simulations showing negative surface temperature anomalies relative to 22 ka until 15 ka over the North Atlantic region (Liu et al., 2009; He, 2011). However, we cannot rule out that the lack of an apparent SSM role in initiating deglaciation may be due to limitations of the relatively coarse resolution GCM used to generate the TRACE ocean temperature field. Between the end of the local LGM and  $\sim 14.6$  ka, the atmospheric temperatures surrounding Iceland for the NROY<sub>tier1</sub> simulations remain below the threshold necessary for substantial surface melt (Fig. 9c, e).



Figure 7. Ice sheet surface elevation (gray shadings) and bedrock elevation (colour shadings) of the mean NROY $_{tier1}$  at 20 ka. Ice surface-height contours are delineated at 450 m intervals, revealing the double-dome configuration. Indicated ice shelf extent (white filled area) was determined from the ensemble-mean ice shelf geometry, excluding grid cells where mean ice thickness was 

**Figure 8.** Basal velocity for one physically self-consistent NROY<sub>tier1</sub> run at 20 ka (run identification number nn10593). (a) The basal velocity field is shown in colour. The background shows the bed topography in greyscale. The streamline ridges are also shown (orange lines) (Benediktsson et al., 2022a). (b) Bedrock topography is shown in colour. Overlain are basal velocity contours at 200 m/yr intervals (black lines), the grounding line (dotted white line), and the ice margin (solid white line).

-

Figure 9. Climate forcing (input) and mass balance components (output) of the IIS over the last deglaciation (19-9 ka) for the NROY<sub>tier1</sub> subset. (a) Global mean eustatic sea level reconstruction. (b) Ocean temperature at 191 m depth from TraCE-21ka deglacial simulation (Liu et al., 2009) for southwestern (red) and northern (blue) grid cells. Gray line represents the mean across all grid cells with  $\pm 1\sigma$  uncertainty (shading). (c) Glacial indices derived from smoothed temperature anomalies (50 years running mean, interpolated to 50 year time steps) from the NGRIP ice core (nor, 2004) and seasonal variations from the LCice model (Geng et al., 2025) (February, April, and July). (d) Mass balance components of grounded ice including net mass balance (blue), surface mass balance (orange dashed), and ice discharge at the grounding line (yellow dashed), with shading indicating uncertainty ( $\pm 1\sigma$ ). (e) Total ice mass (blue, left axis) and normalized mass fluxes (right axis) including surface melt, snow accumulation, calving, and sub-shelf melt. Vertical dashed lines represent the deglaciation phase limits.

•






that the Kolbeinsey ridge ( $\sim 19^{\circ}$ W, 67°N in Fig. 1) may act as a pinning point that buttresses the ice, maintaining continuous and sustained drainage of the ice streams until  $\sim 14.6$  ka. Once this anchoring stops, the ice sheet retreats rapidly across the mid-outer shelf. Such a combination of feedbacks results in the steepest rate of mass loss throughout the deglaciation (Fig. 9e). This aligns with previous geological inferences of marine ice sheet collapse during the Bolling-Allerød (Norðdahl and Ingólfsson, 2015).

Following the phase of rapid mass loss, most of the NROY $_{tier1}$  runs have ice regrowth from 14 to 11.8 ka (III). This is caused by both an increase in accumulation and a substantial reduction in surface melt, which simultaneously increases the surface mass balance (Fig. 9d) and reduces calving by inhibiting hydrofracturing (Fig. 9e). Simulations from the whole ensemble exhibit a range of behaviours over this time interval: there is either a strong decrease in deglaciation rate, stabilization, or a re-advance of the IIS (Fig. A3). Simulations with smaller ice volumes generally exhibit proper re-advance, while those with larger ice volumes experience a decreased deglaciation.

By 11.8 ka (IV), the ice sheet shrinks rapidly. The ice loss is primarily driven by atmospheric warming at the end of the Younger Dryas. This results in a large increase in surface runoff which gives rise to a sharp decrease in surface mass balance as well as a brief increase in calving (Fig. 9d, e). The increased surface melt increases hydrofracturing and therefore calving until it is offset by decreasing the marine fraction of the ice margin.

At 11.5 ka, the IIS margin generally coincides with the present-day coastline, aligning with past empirical inferences based on shorelines and moraines (Norddahl and Pétursson, 2005; Sigfúsdóttir et al., 2018; Benediktsson et al., 2023b). From 11.5 to 9 ka (V), the mass change becomes progressively dominated by the surface mass balance. By 10.7 ka, the magnitude of surface ablation offsets the snow accumulation, resulting in a negative total surface mass balance. The thinning accelerates as surface melt extends to higher elevations (Fig. A9), strengthening the melt-elevation feedback. Once the main ice dome is gone, the residual ice margin retreats gradually to the higher elevations until reaching a configuration comparable to present-day conditions at  $\sim$  9 ka. This is consistent with empirical evidence indicating that ice extent was similar to present prior to 9 ka (Benediktsson et al., 2024).

Our above analysis demonstrates that the IIS deglaciation is dominated by ice discharge during the Phase II rapid retreat episode. Large volumes of ice discharge are associated with the acceleration and expansion of ice streams which drain interior ice to the margin (Fig. 10). This acceleration is initiated by atmospheric warming, which triggers ice shelf disintegration via hydrofracturing (see Sect. 3.2.4 on hydrofracturing below). With the elimination of ice shelf buttressing, ice streams accelerate. Here, ice stream acceleration is therefore triggered by ice shelf removal (see Sect. 3.2.4 on hydrofracturing sensitivity), akin to Antarctica and Greenland (Bell and Seroussi, 2020; King et al., 2020), rather than internal processes as simulated for the Laurentide Ice Sheet during last glacial cycle (Hank and Tarasov, 2024) or IIS throughout MIS 2 (this study). Therefore, contrary to inferences for the Laurentide ice sheet (Stokes et al., 2016), we conclude that ice streams represent unstable entities in changing climate and play a critical role in ice sheet mass balance. However, once the IIS fully retreats onto land (phase V), ice streams largely shut down (Fig. A6), consistent with geomorphological signatures (Aradóttir et al., 2023, 2024).

On the basis of this data-model comparison, we conclude that the IIS deglaciation is most likely caused by a non-linear series of mechanisms. That is, once a temperature threshold is crossed, surface runoff on the ice shelf increases calving via

Figure 10. Ice stream discharge (a) and ice stream discharge normalized by the ice sheet volume (b) of the NROY<sub>tier1</sub> subset. The lines represent the means and the shaded areas the  $2\sigma$  range. Vertical dashed lines represent the deglaciation phase limits as per Fig. 9. Ice stream discharge was calculated over gates placed within bathymetric troughs and valleys.

hydrofracturing, leading to ice shelf disintegration. The resulting elimination of ice shelf buttressing at the grounding line triggers ice stream acceleration and thereby increased ice discharge into the ocean. Hydrofracturing thereby provides a coupling between atmospheric forcing, specifically surface warming and rain, and enhanced grounding line discharge of ice.

## 3.2.4 Sensitivity experiments




To decipher the relative role of hydrofracturing in the IIS deglaciation, we conducted a sensitivity experiment without hydrofracturing during the deglacial interval. The NROY ensemble without hydrofacturing has a much reduced rate of net mass loss during the 14.6 to 14.0 ka phase II interval over which the NROY ensemble with hydro-fracturing loses more than 60% of its mass (Fig. 11). The experiment therefore shows that hydrofracturing plays a key role in the non-linear response of simulated ice sheet retreat to atmospheric warming for at least our NROY set of parameter vectors. With hydrofracturing, the NROY ensemble has an approximate doubling in ice discharge across the grounding line at  $\sim 14.5$  ka, which is also the start of the interval of strongest mass loss from ice calving (Fig. A7). Without hydrofracturing, there is no increase in ice discharge across the grounding line during the whole phase II interval (Fig. A10). The limited calving and sub-shelf melt are not enough to trigger major ice shelf disintegration (Fig. A10) and therefore ice stream acceleration does not occur (Fig. A11). From 15

**Figure 11.** Sea level contribution relative to 15 ka from experiments with hydrofracturing (black) and without hydrofracturing (red), based on the 107 NROY<sub>tier1</sub> parameter vectors. The lines represent the medians and the shaded areas show the  $2\sigma$  ranges.

to 11.6 ka, the no hydrofracturing ensemble has moderate grounding line retreat. Subsequently, at  $\sim 11.6$  ka, once a certain threshold of temperature is crossed, the rate of mass loss increases significantly. Here, contrary to the reference set-up, surface mass balance completely dominates the mass loss, with subshelf melt and, to an even lesser extent, calving playing secondary roles (Fig. A10).

Given the model's high sensitivity to hydrofracturing and its potential reliance on this process to capture critical tier-1 marine constraints, there is the possibility that our analysis is skewed by this reliance. To address this, we reran the whole 2000 member initial ensemble with hydrofracturing turned off over the whole glacial cycle. We then sieved it against the tier-1 data using the same 3σ implausibility threshold applied to NROY<sub>tier1</sub>. Only 2 members meet the NROY criterion, and neither are nominally consistent with the data. In these instances, the model fails to reproduce the last deglaciation dynamics of the IIS, in particular the collapse of the marine ice sheet suggested by the data. As such, for at least the GSM, hydrofracturing is required in order for the IIS simulation to be consistent with our tier I constraints. However, other potential IIS deglaciation drivers not considered here cannot be ruled out, such as episodic volcanic intervals (Lin et al., 2022), or, even more speculatively, catastrophic calving due to large tsunamis.

These results highlight the potentially critical role that hydrofracturing can play in marine ice sheet deglaciation. Pollard et al. (2015) draw similar conclusions in their reconstruction of the Antarctic penultimate deglaciation. We therefore suggest that atmospheric conditions will likely exert stronger influence on future marine ice sheet mass loss than currently projected by models that do not account for hydrofracturing (e.g. Fürst et al., 2016; Reese et al., 2018; Gudmundsson et al., 2019; Payne et al., 2021).

Another potential driver of marine ice sheet collapse is sea level rise. To isolate the relative impact of sea level, we also carried out a sensitivity experiment repeating the 15 ka to 0 ka deglacial interval of the simulations with sea level held constant at its 15 ka value. This was applied to the who NROY set of simulations. This change in forcing had minimal impact on the ensemble. A few simulations had slightly reduced mass loss rates (maximum difference at any time of  $\sim 0.02$  mESL), while the rest had almost no visually discernible response (Fig. A12).

#### 4 Conclusion




We have presented a truncated history matching analysis of the IIS evolution during the last glacial cycle. Out of 2000 simulations, covering a wide range of uncertainties in climatological, oceanographic, glaciological, and solid-Earth parameters, 107 were not ruled out yet by the data and were used to describe the last glacial cycle evolution of the IIS.

Throughout the last glacial period, the IIS fluctuated extensively across the continental shelf with numerous asynchronous ice streams. The ice sheet exhibited an extensive, shallow profile with a double-dome configuration. At the local LGM (23.6-20.9 ka), ice volumes range from 0.41 to 0.76 m e.s.l.. NROY ensemble members include those with an ice bridge between the Greenland and Iceland ice shelves. For a few ensemble members, the ice is grounded across the Denmark Strait. High-resolution sea floor mapping and sediment coring could potentially shed light on such connections and should be explored in future studies.

The IIS remained a relatively stable ice sheet until the Bølling–Allerød, with only minor retreats driven by sea level rise. The rapid deglaciation (14.6-14 ka) was associated with the collapse of the marine-based ice sheet, and started only once discharge significantly outweighed surface mass balance. The mass loss was primarily driven by the atmosphere through disintegration of ice shelves via hydrofracturing, which in turn induces enhanced ice streaming and thereby increased ice flux to the calving margins.

Contrary to past inferences for the Laurentide ice sheet (Stokes et al., 2016) which had extensive terrestrial ice streams as well as more confined marine-based ice streams, our data-constrained numerical experiments indicate that enhanced marine-based ice streaming can drive net mass loss when there is enough surface melt to induce hydro-fracturing of buttressing ice-shelves.

On the basis of the demonstrated pivotal role of hydrofracturing in enabling the model to capture deglacial constraints, we conclude that hydrofracturing is likely a critical mechanism in marine ice sheet deglaciation when the latter are buttressed by ice shelves. As such, we suggest that current projections (e.g. Fürst et al., 2016; Reese et al., 2018; Gudmundsson et al., 2019; Payne et al., 2021) may underestimate the relative contribution of the atmosphere in driving future marine ice sheet mass loss unless they incorporate hydro-fracturing (DeConto and Pollard, 2016; DeConto et al., 2021; Coulon et al., 2024).

Code and data availability. The code is publicly available, and the model is described in a preprint in Tarasov et al. (2025). A code and input data archive for the GSM is hosted on Zenodo (https://doi.org/10.5281/zenodo.14599678, Tarasov, 2025). The IcelandICE database is provided as supplemental Excel (.xlsx) tables.


Author contributions. AAG and LT designed the study. AAG wrote the initial draft, generated Iceland specific GSM inputs, and carried out the experiments and analysis. LT configured the GSM for this context (including sensitivity tests to finalize ensemble parameters), carried out the internal descrepancy assessment, and provided major editorial and experimental design contributions. IOB, JML, and AAG compiled and recalibrated the data. All authors provided feedback on the manuscript.

Competing interests. One author (Lev Tarasov) is a member of the editorial board of Climate of the Past. The authors declare that they otherwise have no conflict of interest.

Acknowledgements. This research has been supported by an NSERC Discovery grant (grant no. RGPIN-2018-06658) and the German Federal Ministry of Education and Research (BMBF) as a Research for Sustainability initiative (FONA) through the PalMod project. Computational resources were provided by ACEnet and grants from the Canadian Foundation for Innovation. We thank Audrey Parnell, Kevin Hank, and Matthew Drew for constructive reviews of a draft. We also thank Anne de Vernal for insightful discussion on the potential ice bridge connecting the Greenland and Iceland ice.

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

# **Appendix A: Supplementary Figures**

**Figure A1.** Iceland observational constraint database with site locations and identification numbers for past ice thickness data (paleoH), and past ice extent data (paleoExt).

.

Figure A2. Past ice thickness misfit scores (absence of ice). The data ID locations are shown in Fig. A1.

.

**Figure A3.** Time series of IIS area during the last glacial cycle. Initial full ensemble (gray shaded area) and sieved non-ruled out yet (NROY<sub>tier1</sub>) sub-ensemble (blue distribution) with median NROY<sub>tier1</sub> (black curve).

0.6 0.0350.03 0.5 0.025 mean volume (msle) variance normalized by the mean volume 0.4 0.02 0.3 0.015 0.2 0.01 0.1 0.005 0 -120 -110 -100 -90 -80 -70 -60 -50 -40 -30 -20 -10 0 Time (ka)

**Figure A4.** Time series of IIS mean volume (left axis) and variance normalized by the mean volume (right axis) during the last glacial cycle for the NROY<sub>tier1</sub> subset.

٠

Figure A5. Basal velocity at six ice stream gates of the IIS for 5 NROY<sub>tier1</sub> runs.

Figure A6. Basal velocity fields of the IIS for 4 NROY $_{tier1}$  runs (run identification number nn10593, nn10114, nn1184, and nn1468) at different times.

Figure A7. Climate forcing (input) and mass balance components (output) of the IIS over the last glacial cycle (120-0 ka) for the NROY<sub>tier1</sub> subset. (a) Global mean eustatic sea level reconstruction. (b) Ocean temperature at 191 m depth from TraCE-21ka deglacial simulation (Liu et al., 2009), with the gray shading indicating uncertainty ( $\pm 1\sigma$ ) around the mean (gray line). (c) Glacial indices derived from smoothed temperature anomalies (50 years running mean, interpolated to 50 year time steps) from the NGRIP ice core (nor, 2004) and seasonal variations from the LCice model (Geng et al., 2025) (February, April, and July). (d) Mass balance components of grounded ice including net mass balance (blue), surface mass balance (SMB, orange dashed), and ice discharge at the grounding line (yellow dashed), with shading indicating uncertainty ( $\pm 1\sigma$ ). (e) Total ice mass (blue, left axis) and normalized mass fluxes (right axis) including surface melt, snow accumulation, calving, and sub-shelf melt.

Figure A8. Time series of the ice shelf area during the last deglaciation for the NROY<sub>tier1</sub> subset. The line represents the means and the shaded area the  $2\sigma$  range.

.

Figure A9. Surface and bedrock elevation at the center of the IIS for the NROY<sub>tier1</sub> subset. The lines represent the means and the shaded areas the  $2\sigma$  range.

٠

**Figure A10.** Mass balance components (output) of the IIS from 15 to 9 ka for experiments without hydrofracturing. (a) Mass balance components of grounded ice including net mass balance (blue), surface mass balance (SMB, orange dashed), and ice discharge at the grounding line (yellow dashed), with shading indicating uncertainty ( $\pm 1\sigma$ ). (b) Total ice mass (blue, left axis) and mass fluxes normalized by the ice sheet total mass (right axis).

•

Figure A11. Ice stream discharge (a) and ice stream discharge normalized by the ice sheet volume (b) of the tested subset without hydrofracturing. The lines represent the means and the shaded areas the  $2\sigma$  range. Ice stream discharge was calculated over gates placed within bathymetric troughs and valleys.

.

39

**Figure A12.** Sea level contribution relative to 15 ka from experiments with sea level change (black) and without sea level change (red) for 2 NROY<sub>tier1</sub> parameter vectors (run identification number nn1131, and nn1395, the latter showing the strongest sensitivity in the ensemble).

.