# Peer review of "Growth and decay of the Iceland Ice Sheet through the last glacial cycle"

_EGUsphere, 2025_

## Author Comment (AC1)

**Author's response to Anonymous Referee 1**

January 27, 2026

**General comments**

This is a well written and timely paper that uses numerical modelling constrained by palaeo-glaciological and geochronological data to produce a set of ice sheet histories for the Icelandic Ice Sheet through the last glacial cycle. I think the subject of the paper is suitable for Climate of the Past and could be published subject to some minor-moderate revision that address the following points:

Dear reviewer 1,

Thank you very much for your thoughtful and constructive comments. We provide a point-by-point response below. The reviewer comments are shown in blue and our replies in black.

**Specific comments**

1. The interpretation that hydrofracturing drove ice shelf disintegration and thus collapse, clearly relies upon the presence of ice shelves fringing the ice sheet during deglaciation. But what is the direct geological evidence that ice shelves did indeed fringe the IIS during retreat? Rather than an ice shelf could the retreating ice sheet margin have been in the form of a grounded tidewater margin? Can the authors rule out the latter scenario and, if not, would this have an impact on their interpretation of the role of hydrofracturing? There needs to be a more explicit justification for the existence of ice shelves based on marine geological data. I am not saying I disagree with the authors but rather that this is important to their study and they need to provide a more convincing justification for ice shelves presumably from published geological data. This could be included within #2 below.

   This is a good question and points to a limitation in our original analysis in our failure to isolate ice shelf and tidewater margins. We will expand the discussion on ice shelf evidence and tidewater margins in the revised manuscript.

To our knowledge, direct geological evidence for ice shelves dating from the last deglaciation does not currently exist, though this study motivates future coring in locations where our model simulates ice shelves at the onset of MWP-1A.

Figure 7 in the manuscript may overrepresent the actual ice shelf area, as it shows the ensemble mean (although grid cells for which mean ice thickness was $< 25$ m were excluded). Individual ensemble members exhibit a mix of ice shelves and grounded tidewater margins. Overall, the northern margin is mostly bounded by extensive ice shelves, while the southern margin is tidewater at 15 ka for the majority of NROY simulations (see Figs. 1 and 2 below).

Hydrofracturing is also a key component of tidewater calving in the GSM [c.f. equations 31 and 32 in Tarasov et al., 2025]). Thereby, to disentangle the relative contributions of tidewater versus ice shelf hydrofracturing, we performed sensitivity experiments isolating each mechanism. We find that they both contribute comparably to total hydrofracturing (see Fig. 3 below). Thus, regardless of whether margins terminate in ice shelves or grounded tidewater glaciers, hydrofracturing remains critical in deglaciation.

[Figure]

Figure 1: Basal velocity fields of the IIS at 15 ka for 4 NROY$_{tier1}$ runs (run identification number nn1864, nn1490, nn1526, and nn1468). The present-day coast line is shown in white and the groundling line in magenta.

[Figure]

Figure 2: Density distribution of floating ice within the NROY$_{tier1}$ sub-ensemble at 15 ka. The -860 m contour is represented by the black dotted line.

[Figure]

Figure 3: Sea level contribution relative to 15 ka from experiments with hydrofracturing (black), without hydrofracturing (red), without hydrofracturing only for tidewater margins (orange), and without hydrofracturing only for ice shelf (blue) based on the 107 NROY$_{tier1}$ parameter vectors. The lines represent the medians and the shaded areas show the 2-sigma ranges.

.

2. I think it would be helpful if the authors included a summary of the key points from the geological data of the ice sheet history – perhaps summarised by different sector and noting in particular the geochronology on retreat timing and rate. This could be an additional section or included in the Introduction. It does not have to be very long but it would be helpful. At the moment there is only a section on 'Palaeo-constraints which appears as section 2.2 in Methods.

We agree. We will include a summary of the empirically-inferred full last glacial cycle ice sheet evolution.

3. At the very end of the paper (lines 424-428) sea level rise is mentioned as a potential driver of marine ice sheet collapse. The authors state that to address the impact of sea level they carried out a sensitivity experiment with sea level held constant at its 15 ka value. This had little impact. How realistic is holding sea level constant given that the sea level jump associated with Meltwater-Pulse 1A occurred shortly after this time and was broadly coincident with the timing of the rapid deglaciation and collapse of marine-based ice at 14.6-14.0 ka?

The point of a sensitivity experiment is to isolate a physical process, in this case, the relative role of deglacial sea-level rise in driving ice sheet collapse by exclusion of such a rise. Such isolation will generally entail imposition of idealized (i.e., contrary to

paleo records) boundary conditions or forcings.

4. Line 27 – presumably reconstructing IIS evolution is also challenging due to the absence or sparsity of geochronological control?

The subsequent paragraph addressed the limited constraint from proxies, but your point makes clear the current statement is misleading. As such, we have revised the beginning of the offending paragraph to:

Reconstructing IIS evolution is challenging due to limited empirical constraints and large process uncertainties. The latter includes: (i) oceanic and atmospheric...

**References**

L. Tarasov, B. S. Lecavalier, K. Hank, and D. Pollard. The glacial systems model (gsm) version 25g. *Geoscientific Model Development*, 18(23):9565–9603, 2025. doi: 10.5194/gmd-18-9565-2025. URL https://gmd.copernicus.org/articles/18/9565/2025/.